Fracture strength of cad-cam milled polyetheretherketone (PEEK) post-cores vs conventional post-cores; an in vitro study

Direk Ayetullah 1
Tekin Samet 2
Khurshid Zohaib zsultan@kfu.edu.sa 3 4
1 Prosthodontics Clinic, Batman Oral and Dental Health Center , Batman , Turkey
2 Department of Prosthodontics, Faculty of Dentistry, Firat University , Elazig , Turkey
3 Department of Prosthodontics and Dental Implantology, King Faisal University , Al Ahsa , Saudi Arabia
4 Center of Excellence for Regenerative Dentistry, Department of Anatomy, Faculty of Dentistry, Chulalongkorn University , Bangkok , Thailand
Uversky Vladimir
Electronic publication date: 2024 Sep 5
Publication date: 2024
Volume: 12
Electronic Location ID: e18012
Received 2024 Mar 8; Accepted 2024 Aug 9
Copyright: ©2024 Direk et al.
Copyright year: 2024
Copyright holder: Direk et al.
License: This is an open access article distributed under the terms of the Creative Commons Attribution License, which permits unrestricted use, distribution, reproduction and adaptation in any medium and for any purpose provided that it is properly attributed. For attribution, the original author(s), title, publication source (PeerJ) and either DOI or URL of the article must be cited.
License URL: https://creativecommons.org/licenses/by/4.0/

Keywords: Fixed prosthodontics, Polyetheretherketone, Zirconia, Fracture strength, Post-core, CAD-CAM

Funding: Deputyship for Research & Innovation, Ministry of Education in Saudi Arabia through the project number INST218 The authors received support from the Deputyship for Research & Innovation, Ministry of Education in Saudi Arabia through the project number INST218. The funders had no role in study design, data collection and analysis, decision to publish, or preparation of the manuscript.

==============================
Background

The aim of this study was to compare the fracture strength and fracture modes of post-cores produced with CAD-CAM from modified polyetheretherketone (PEEK) materials with other custom-produced post-cores.

Methods

Sixty human mandibular first premolars with equal root sizes were used. The teeth were divided into six groups (n = 10), and root canal treatment was performed. The teeth were separated from the roots over 2 mm from the cemento-enamel junction. As a result of the decoronation process, a 1 mm wide shoulder line was obtained for all teeth. For the fracture strength test, 10 mm deep post spaces were created on the teeth with a 1.6 mm diameter driller. Post-core groups consisted: everStick® glass fiber post-core (Group GF), zirconia post-core (Group Z), metal (Cr-Co) post-core (Group M), PEEK post-core without filler (Group UP), PEEK post-core with 20% TiO2 Filler (Group TP), and post-core with 20% ceramic filler (Group CP). Following the application of posts to post spaces, copings were created and cemented on the samples. With the universal tester, a force was applied to the long axis of the tooth with a slope of 135°. The mean fracture strength (N) between the groups was statistically evaluated using one-way ANOVA, and pairwise mean differences were detected using post hoc Tukey’s HSD test among the groups.

Results

According to the results of the statistical analysis, a significant difference was found between the groups in terms of mean fracture resistance (p < 0.05). Group Z (409.34 ± 45.72) was significantly higher than Group UP (286.64 ± 37.79), CP (298.00 ± 72.30), and TP (280.08 ± 67.83). Group M (376.17 ± 73.28) was significantly higher than Group UP (286.64 ± 37.79) and Group TP (280.08 ± 67.83). There were no statistically significant differences between the means of the other groups (p > 0.05). Among all the groups, Group Z exhibited a higher prevalence of repairable failure modes, while the rest of the groups predominantly experienced irreparable failure modes.

Conclusion

In our study, zirconia and metal post-core samples showed higher average fracture strength values than PEEK post-cores groups. Repairable failure modes were more common in the zirconia post-cores, whereas the opposite was observed in the other groups. Further experimental and clinical trial studies are needed before PEEK materials can be used as post materials in the clinic.

Introduction

Prosthetic rehabilitation of damaged teeth is complicated and highly requires skilled specialist with comprehensive treatment planning. In the past, different techniques and materials were used in these cases. It is important that the post system does not cause stress to the tooth under occlusal forces. In addition, when the posts are placed and the treatment needs to be repeated, the post should be comfortably removed from the tooth without excessive preparation (Mahmoudi et al., 2012; Uctasli et al., 2021; Fráter et al., 2021).

Metal post-cores increase the possibility of irreparable fractures in the remaining tooth structure. The cast metal and metal prefabricated posts used today cannot fully meet this requirement owing to their high elasticity modules (Barcellos et al., 2013). In addition, aesthetic posts are a good alternative to metal posts in post-core treatment (Akkayan & Gülmez, 2002). However, although color harmony and aesthetic expectations are met, these posts have several disadvantages. One of the main disadvantages of glass fiber posts is their industrial prefabrication process. Owing to the diversity of commercial presentations, post diameters are standardized and may not adequately fit into the root canal cavity. This requires further widening of the root canal for post placement or opting for a smaller-diameter post (Gutiérrez, Guerrero & Baldion, 2022). It has been shown that glass-ceramic restorations are vulnerable to lateral forces, aluminum oxide-reinforced ceramics take a long time and show technical sensitivity, and zirconium oxide-based prefabricated posts have low fracture resistance compared to metal posts and are difficult to remove from the root canal after failure (Morgano & Brackett, 1999).

The modulus of elasticity difference between metal alloys and dentin causes stress concentration around the post and irreparable root fractures (Al-Omiri et al., 2010). Therefore, various post-core materials have been investigated to ensure long-term safety (Akkayan & Gülmez, 2002; Fraga et al., 1998; Newman et al., 2003).

Previous studies have reported that more appropriate stress distributions occur when post materials with a lower modulus of elasticity, such as glass fibers, are used (Okada et al., 2008; Beh, Halim & Ariffin, 2023). However, because glass fiber-supported posts are usually prefabricated, they are limited in their suitability for root canal shape. In addition, although glass fiber-reinforced posts have lower modulus of elasticity 45.7 to 53.8 GPa than metal alloy posts, which are approximately three times the modulus of elasticity of dentin 18.6 GPa (Cheleux & Sharrock, 2009; Craig & Peyton, 1958).

The formable fiber posts are highly flexible posts with a resin-impregnated non-polymerized glass fiber structure. It consists of intertwined polymer networks of unevenly distributed fiber bundles with sizes between 0.9 and 1.5 mm saturated with polymethylmethacrylate (PMMA) and bis-GMA. This increases the bonding between the post and resin cement, reducing adhesive failure and microleakage. Custom impregnated with resin fiber posts can be shaped by adapting the shape of the root canal. Similar to the lateral condensation technique, additional strips can be added to fill the root canal. The fibers in the coronal part can be expanded by opening to provide a better core connection (Alnaqbi, Elbishari & Elsubeihi, 2018; Doshi et al., 2018). This allows these posts to be used in wide, oval, or curved root canals. It also increases the fracture resistance of the tooth root. Because its flexural strength and elasticity are similar to those of dentin, it allows incoming stresses to be transmitted throughout the root (Doshi et al., 2018). Polyetheretherketone (PEEK) post-cores used in dentistry have been the subject of research with the claim that they are an alternative with properties that will eliminate the disadvantages of many posts used so far (Tekin et al., 2020).

PEEK, a synthetic polymer-based biomaterial, has been a mainstay in orthopedics for many years and has also gained prominence in dentistry over the past decade. Well-known for its physical and mechanical properties, which closely resemble those of bone and dentin, as well as its similarity to natural tooth color, PEEK has emerged as a preferred material. It is utilized in various dental applications including dental implants material, crown and bridge construction, and removable prosthesis fabrication (Najeeb et al., 2016). In addition, because it can be connected with PEEK resin materials, prostheses with better aesthetics can be obtained by veneer treatment with resin composites (Uhrenbacher et al., 2014). PEEK is an alternative that can be produced by computer-aided design and computer-aided manufacturing (CAD-CAM) (Stawarczyk et al., 2015), lost-wax technique (Zoidis, Papathanasiou & Polyzois, 2016) and 3D printing (Wang & Zou, 2022; Challa et al., 2022). Owing to their acceptable fracture resistance, better stress distribution, and shock absorption properties, high-performance polymers have been recognized as dental materials as alternatives to metals and glass-ceramics. Although unfilled PEEK has a lower modulus of elasticity (approximately 4 GPa) than dentin, PEEK has been reported to exhibit high compressive strength (plastic deformation of approximately 1200 N (Stawarczyk et al., 2014). All these properties and biocompatibility, as well as its wide manufacturing and processing capabilities, make PEEK an attractive dental material for the production of customized post-core systems (Kasem, Shams & Tribst, 2022).

Although PEEK posts do not have as high fracture strength as metal posts, they exhibit good mechanical performance competitive with nano-ceramic composite and glass fiber posts and have a higher incidence of repairable failure modes (Teixeira et al., 2020). Custom made PEEK posts have been shown to exhibit higher fracture strength compared to custom glass fiber posts regardless of different manufacturing techniques (milled and pressed) (Abdelmohsen et al., 2022).

This study aimed to compare the fracture strength and fracture modes of post-cores produced with CAD-CAM from modified PEEK materials with other custom-produced post-cores. The null hypothesis of this study is that there is no significant difference between the fracture strength and failure modes of different post-core materials.

MATERIALS & METHODS

Ethical approval was obtained from the Firat University Non-Interventional Research Ethics Committee (session number 2020/08-32). Written consent was obtained from eligible patients whose mandibular premolars were planned to be extracted due to advanced periodontal disease and orthodontic treatment.

Sample preparation

The sample size was calculated using a computer program (G*Power software; Heinrich Heine University, Düsseldorf, Germany). Samples were randomly distributed to six groups. This study consisted of sixty single-root and single-canal labiolingual and mesiodistal dimensions of the same (± 0.2) mandibular premolars with the same root length (15 ± 0.5) were preferred. The reason is other teeth in the dental arch, such as molars or anterior teeth, may lead to different results. Premolars are commonly extracted teeth in dental clinics for periodontal and orthodontic reasons and they have a single root (Kim et al., 2017). Teeth with decay, cracks, crown destruction, or endodontic treatment were not included in the study. Radiographic images were taken from all teeth in both the mesiodistal and buccolingual directions. The teeth were stored in a 0.5% chloramine T solution for 5 min and then teeth were stored in a 0.5% chloramine T solution at 4 °C glass containers. The tissue residues on the root surfaces of the teeth were removed using periodontal instruments.

The crowns of the teeth were removed under water cooling from the 2 mm coronal cemento-enamel junction. As a result of the decoronation process, shoulder finish line preparations with diamond burs were created on teeth with a width and depth of 1 mm.

Drills of sizes 1, 2, and 3 were used for coronal preparation. The working length was 1 mm shorter than the length of the root canal. For root canal preparation of the teeth, apical #40 ProTaper was prepared using rotary files (Dentsply Maillefer, Ballaigues, Switzerland). Two milliliters of 2.5% sodium hypochlorite (NaOCl) irrigation solution were used for each file change. Irrigation with 5 ml of 17% ethylenediaminetetraacetic acid (EDTA) and 2.5% NaOCl was completed 1 min after preparation. Finally, the canals were washed with 2 ml saline and dried using paper points (Dentsply, Tulsa Dental, Tulsa, England). The teeth for which root canal preparation was completed were filled with AH Plus canal paste with resin content (Dentsply, Dentsply DeTrey, Konstanz, Germany) and gutta-percha cones (Dentsply DeTrey, Konstanz, Germany) using the lateral condensation technique. The root canal openings were closed with temporary filling material (Cavit G; 3M ESPE, Seefeld, Germany) and kept at 37 °C in containers with 100% humidity for one-week for simulate the condition of the oral cavity during the setting of the root canal filling. Post space was prepared using peeso reamers up to 10 mm length until four mm of gutta-percha remained in the apical area.

The production of the posts, preparation of the samples and mechanical tests were performed. The materials used for the production of the post-core samples in this study are listed in Table 1. (n = 10)

Table 1 Materials used in the production of post-cores in the study.

	Group	Manufacturer	
Glass Fiber Post	Group GF	everStick®Post; StickTech Ltd., Turku, Finland	
Zirconia Post	Group Z	Upcera CT Color; Shenzhen, Guangdong, Çhina	
Metal Post (Cr-Co)	Group M	Coprabond K, WhitePeaks, Essen, Germany	
Unfilled PEEK Post	Group UP	Juvora; Juvora Ldt, Thornton Cleveleys, Lancashire, England	
Modified with 20% TiO2 PEEK Post	Group TP	KERA®starPEEK; Eisenbacher Dentalwaren ED GmbH, Wörth am Main, Germany	
Modified with 20% Ceramic PEEK Post	Group CP	breCAM.BioHPP; Bredent GmbH, Senden, Germany	

Preparation of EverStick (E-glass) post and resin composite core samples

A total of 1.5 mm diameter glass fiber posts (everStick Post, Stick®Tech, Turku, Finland) were selected for the group GF. The post was advanced to its working length. They were placed in the channel in one piece, together with additional post parts cut to appropriate dimensions for the gaps between the post and dentin, and the samples were placed in a closed box.

The shaped everstick posts were then removed from the root canal and polymerized with light for 40 s according to the manufacturer’s instructions. For cementation, an adhesive primer consisting of two bottles of liquid in the Panavia F2.0 Complete Kit (ED Primer II Liquid A-B; Kuraray, New York, NY, USA) was applied to the post cavity for 30 s, and dual-curing adhesive resin cement (Panavia F2.0, Kuraray, Noritake, Kurashiki, Japan) was coated around the post and gently inserted into the channel to avoid air gaps. The excess cement was removed using a cotton brush. Continuous finger pressure was applied to hold the posts in place. Polymerization was then performed using LED light (Valo Cordless LED Curing Light; Ultradent, South Jordan, UT, USA) from different angles for 40 s. After cementation, the adhesive primer (ED Primer II; Kuraray, New York, NY, USA) in the Panavia F2.0 Complete Kit was applied for 30 s, and the specimens were light-cured for 20 s. A conventional resin composite (Clearfil Majesty; Kuraray Medical, Tokyo, Japan) was used to form four mm cervico-incisal sized cores and light-cured.

Preparation of zirconia post-core samples with CAD/CAM

Measurements of the post space were obtained using polyvinyl siloxane impression material (Elite HD+; Zhermack, Rovigo, Italy), with impression posts formed in sizes suitable for the existing post spaces.

A powder (CEREC Optispray, Dentsply Sirona) was applied to the outer surfaces of the impressions for a more perfect transfer to the computer environment. All impressions were scanned and digitized using a computer-assisted scanning device (inEos X5; Dentsply; Sirona, Bensheim, Germany). Cores with cervicoincisal heights of 4 mm were designed for each model. Three-dimensional post models created in a digital environment were designed using computer-aided design software (inLab SW 20.0 Dentsply; Sirona, Bensheim, Germany). The thickness of the cement was determined to be 80 µm. Zirconia blocks (Upcera) were produced using a computer-aided manufacturing device (CEREC inLab MC X5; Dentsply Sirona, Bensheim, Germany) with a milling pitch of 98.5 * T16 mm. The preparation of the post spaces and the design of the post-cores were the same for all groups fabricated using the CAD/CAM technique.

Zirconia post-cores were roughened with 50 µm Al2 O3 (Blastmate II; Ney, Yucaipa, CA, USA) particles. ED Primer II in the Panavia F2.0 Complete Kit was applied to the enamel and dentin surfaces, and alloy primer was applied to the zirconia post-cores. For cementation, dual-cure resin cement was applied under finger pressure and cured with an LED light device from different angles for 40 s. The excess cement was removed using a cotton brush. Continuous finger pressure was applied to prevent the displacement of the placed post.

Preparation of metal (Cr-Co) post-core samples with CAD/CAM

Post-cores were obtained by milling the Co-Cr block (Co 61%, Cr 27.9%, W 8.56%, Mn 0.23%, Fe 0.11%, Si 1.73%, C 0.07%; Coprabond K, WhitePeaks, Essen, Germany) with CAD/CAM. ED Primer II was applied to the enamel and dentin surfaces and Alloy Primer was applied to the metal post surface after 50 µm Al2O3 sandblasting and the post-cores were cemented.

Preparation of PEEK post-core samples with CAD/CAM

Post-cores were obtained from three different PEEK discs using CAD/CAM (Table 1). The PEEK post-cores were sandblasted with 50 µm Al2O3 particles at a distance of 10 mm from the surface for 15 s at a pressure of 0.4 MPa. Thus, rough surfaces were obtained, on which the adhesive resin cement could adhere better.

The specimens were washed with distilled water for 60 s and dried using oil-free compressed air. Visio.link (Bredent, Senden, Germany) was applied to the post surface, ED Primer II was applied to the enamel and dentin, and cementation of the post-cores was performed similarly to the other CAD/CAM groups.

After the preparation of the specimens for all test groups and the cementation of the post-cores was completed, the specimens were scanned with a digital scanner (Hint-Els Scanner; Hint-Els DentaCAD Systeme, Germany) and the design process (DWOS software, Dental Wings Inc., Montreal, Canada) was performed. The computerized data were transferred to the production department, and metal crowns were prepared from Cr-Co metal alloy powder (SINTTECH, Clermont-Ferrand, France) with a laser metal sintering device (EOSINT M280; EOS GmbH Electro Optical Systems, Germany). The crowns were cemented to the teeth using a glass ionomer cement (Meron; Voco, Cuxhaven, Germany).

The restored specimens were stored in distilled water at 37 °C for one week. The root surfaces of the prepared specimens were coated with a viscous polyvinylsiloxane material (Elite HD+; Zhermack, Rovigo, Italy) approximately 0.25 mm thick. The specimens were embedded in 20 × 20 × 20 mm plastic molds with acrylic resin parallel to the long axes of the teeth.

Thermocycling and fracture test

After cementation, the samples were stored at 37 °C and 100% humidity for 24 h before thermal cycling. The prepared test specimens were subjected to thermal cycling 6,000 times between 5°–55 °C with 30 s immersion time and 2 s transfer time.

The specimens were placed in the universal test device (Instron 3345, Instron Corporation, Norwood, MA, USA) set at an inclined plane at 135° to simulate oblique forces on teeth in the oral environment.

The fracture test was performed by applying a force from 2-mm below the tubercle apex of the lingual surface of the buccal tubercles of the crowns to the lingual surface of the crowns at a crosshead speed of 1.0 mm/min with a one mm thick force-applier tip of the universal tester until the specimens fractured. The forces causing fracture were recorded in Newtons (N) (Fig. 1)

Figure 1 Positioning of the specimen before the fracture strength test.

Source credit: Ayetullah Direk.

The fracture modes formed after the fracture were examined with a stereomicroscope at x20 magnification and classified according to restorability. The fracture modes were classified according to the location of failure as follows: mode 1, coronal fracture (displacement of the prosthetic set with fracture at the cementation line); mode 2, coronal third of the root; mode 3, middle third of the root; mode 4, apical third of the root; and mode 5, vertical root fractures. Modes 1 and 2 describe the repairable failure pattern and modes 3–5 describe the irreparable failure pattern (Barcellos et al., 2013). Schematic illustration of the failure mode classification was shown in Fig. 2. The lines represent the apical boundary of failure.

Figure 2 Schematic illustration of the failure mode classification.

The lines represent the apical boundary of failure. Photo credit: Pinar Cetinkaya.

Statistical analysis

The forces leading to fracture were statistically analyzed by one-way analysis of variance (ANOVA). Tukey’s honest significant difference (HSD) test, among the post hoc tests, was used to identify which groups differed with SPSS 23.0 (SPSS Inc., Chicago, IL, USA). Statistical significance was set at p < 0.05 level. The flowchart describing the design of the study was shown in Fig. 3.

Figure 3 Flowchart describing the design of the study.

RESULTS

Table 2 shows fracture strength results (mean ± standard deviation) of experimental test groups after static loading and minimum-maximum values in N. According to the one-way analysis of variance, the mean fracture strength values were found to be statistically significantly different between the groups (p < 0.05). Tukey’s HSD test was used to conduct pairwise comparison of means across six groups.

Table 2 Mean, standard deviation, maximum and minimum force values (N) for each groups.

Different letters indicate significant differences in composition between materials (Tukey’s test, p < 0.05).

Groups	n	Mean ± SD	Minimum	Maximum	
Z	10	409.34 ± 45.72	318.63	475.55	
M	10	376.17 ± 73.28	301.22	538.83	
GF	10	347.81 ± 53.54	301.17	452.25	
UP	10	286.64 ± 37.79ab	213.66	352.22	
CP	10	298.00 ± 72.30a	225.74	477.95	
TP	10	280.08 ± 67.83ab	187.98	436.71	
Total	60	333.00 ± 75.37	187.98	538.83	
Notes.

SD, Standard deviation.

a Statistically significant difference according to Z group.

b Statistically significant difference according to M group.

No statistically significant difference was found between the means of the other groups.

The mean of the Group Z (409.34 ± 45.72) was significantly higher than that of the UP (286.64 ± 37.79), CP (298.00 ± 72.30), and TP (280.08 ± 67.83). The mean of the Group M (376.17SD) was significantly higher than that of the UP (286.64 ± 37.79) and TP (280.08 ± 67.83). There were no statistically significant differences between the mean values of the other groups (p > 0.05). The details of multiple comparisons between the groups are presented in Table 2.

The failure modes of the specimens in the present study as a result of static loading are listed in Table 3. Most of the repairable fracture modes were observed in specimens restored with zirconia posts.

Table 3 Failure modes of specimens.

	Group GF	Group Z	Group M	Group UP	Group TP	Group CP	
Mode 1	–	2	–	–	–	–	
Mode 2	1	6	–	–	3	4	
Mode 3	7	2	9	8	5	5	
Mode 4	–	–	–	2	2	1	
Mode 5	2	–	1	–	–	–	

DISCUSSION

This study aimed to evaluate the fracture strength and fracture modes of post-core restorations made of different materials in endodontically treated teeth. Post-core restorations were fabricated from glass fiber, zirconia, CAD/CAM milled metal (Cr-Co), CAD/CAM milled PEEK without filler, CAD/CAM milled PEEK with 20% ceramic content and CAD/CAM milled PEEK with 20% TiO2, which are used in various fields of dentistry. According to the results of this study, our null hypothesis that there would be no difference between the fracture strengths of teeth restored with post-cores made of different materials after endodontic treatment was rejected. A statistically significant difference was found between the mean values of the fracture strength.

The advancement in digital devices in dentistry and new ceramic based materials in the form of powder or blocks with superior properties allows the preparation of aesthetic monolithic restorations. Post-core restorations prepared as a single piece compensate for the missing structure at the post-core interface and reduce treatment failure (Liu, XL & Wang, 2010). The industrial processing of blocks provides a higher structural reliability (Beuer, Schweiger & Edelhoff, 2008). In addition, this process allows the design of restorations using software that allows better control of the resin cement gap, as well as the thickness, shape, marginal features, and characteristics of the restoration (De Andrade et al., 2019). Based on these advantages of the CAD/CAM method, in this study, post-cores were milled from PEEK, zirconia, and metal (Cr-Co) blocks using CAD/CAM. The post fabrication process included scanning and reproducing the post using the CAD/CAM system so standardization of size and shape of the posts was aimed in the test materials. The success of dental treatment is directly related to the strength of the materials and the region where they are used. In healthy young adults, occlusal forces in the mandibular first premolar region have been reported to be approximately 178 N in females and 254 N in males. The fracture strength values obtained in our study were higher than these values.

In a recent in vitro study, fracture resistance and fracture modes of PEEK, nanoceramic composites, Ni-Cr and glass fiber nano hybrid composite post-core restorations were investigated. The Ni-Cr post-core group exhibited the highest fracture strength. Regarding the failure modes, 91.7% of the Ni-Cr post-core samples showed non-restorable fractures, while the glass fiber post group had a 91.7% restorable failure mode. A rate of 83.3% for PEEK and 66.7% for nanoceramic composite post-cores exhibited restorable fracture modes (Ferrario et al., 2004; Teixeira et al., 2020). In our study, metal post and zirconia post-core specimens with a high modulus of elasticity showed the highest fracture strength values, similar to a previous study. However, in our study of the fracture modes, only in the zirconia post-core group, the repairable fracture mode was greater than the irreparable mode.

In the previous study, different results have been obtained regarding fractures in different parts of the root caused by posts. Mannocci et al. reported that fiber-reinforced posts minimized the risk of root fracture compared with zirconia posts (Mannocci et al., 2002). When the failure modes of teeth restored with rigid posts were examined, failures usually occurred in the apical half of the root. However, in a study by Bittner, Hill & Randi (2010) with fiber-reinforced posts, fractures were mostly irreversible and occurred in the cervical half of the root. This result reversed the view that posts with a modulus of elasticity similar to that of dentin generally produce a more favorable failure pattern (Bittner, Hill & Randi, 2010). In the present study, root fractures were generally observed in the cervical third region. This finding is partly supported by the results of Bittner, Hill & Randi (2010). Similarly, repairable fractures occurred more frequently in the zirconia post-core groups, whereas irreparable fractures occurred mostly in the glass fiber post group.

The evaluation of the literature generally revealed the fact that higher fracture resistance was found for zirconia posts when compared with fiber posts (Beck et al., 2010). However, in a recent study it was observed that glass-fiber post specimens were more successful in terms of fracture strength than zirconia and PEEK post-core. According to the restorability of the specimens, PEEK post-core samples showed more restorable fractures than glass fiber and zirconia samples (Ozarslan, Buyukkaplan & Ozarslan, 2021). However, zirconia and glass fiber post-cores showed higher fracture strength than PEEK posts, similar to the results obtained from the group (1.6-mm-diameter: 10 mm) most similar to the post sizes used in the study by Ozarslan, Buyukkaplan & Ozarslan (2021). According to the restorability of the fractured specimens in the present study, zirconia post-core samples showed a more restorable fracture mode than PEEK post-core samples and glass fiber post-core samples. This difference may be due to the fact that the posts produced in our study were milled together with the core structures, and the specimens from the zirconia post group mostly fractured with a favorable failure mode in the area adjacent to the core border of the posts.

Haralur (2021) evaluated specimens restored with fiber-reinforced composite posts, PEEK posts, and polymer-infiltrated ceramic posts in terms of fracture toughness and fracture modes. PEEK posts showed better fracture strength results than the other two groups. While the number of unrepairable fracture specimens was higher in the PEEK post group, the number of repairable specimens was higher in the other two groups (Haralur, 2021). In our study, the Everstick glass fiber post-cores had a higher fracture strength than the PEEK post-cores. The results of the present study are in agreement with the previous study with regard to the number of unrepairable fracture specimens was higher in the PEEK post-core groups. In our study, the fracture strength values of our study were also lower than the previous study. The reason for this difference may be the weakening of the amount of remaining dentin due to the wider post thickness in our study.

A previous study reported that the fracture resistance of PEEK posts alone was lower than glass fiber posts and they were not suitable as a post material (Sugano et al., 2020). Although the fracture strength of the custom-shaped glass fiber posts was higher than the PEEK posts, no statistically significant difference was found in our study. These results were similar to those of a study by Lima, Ferretti & Caldas (2023). In a study evaluating the fracture strength of custom-made ceramic-containing PEEK posts, similar to our study, the mean fracture strength of ceramic-containing PEEK posts (298.00 ± 72.30) was very close to the results of the previous study (286.16 ± 67.09 N). In addition, similar to both studies, the mean fracture strengths of glass fiber posts were higher than PEEK posts, which exhibited the lowest fracture strength value among all groups (Fathey, Azer & Abdelraheem, 2024).

Conflicting hypotheses regarding the modulus of elasticity of endodontic posts have been presented by researchers. Previous studies have suggested the use of posts with high modulus of elasticity to reduce stress values (Yaman, Karacaer & Sahin, 2004; Asmussen, Peutzfeldt & Sahafi, 2005). According to these researchers, stiffer posts increase flexural resistance, resulting in less deformation during transverse loading. Nakamura et al. (2006) recommended the use of posts with a modulus of elasticity close to dentin to form a homogeneous whole with dentin and increase biomechanical performance. In our study, the number of repairable fracture specimens was higher in zirconia post-cores, which have a higher modulus of elasticity than dentin. The number of irreparable fracture specimens was higher than the number of repairable specimens in metal (Cr-Co) with a high modulus of elasticity and PEEK posts with a low modulus of elasticity. Although the failure mode in the metal post group was described as an irreparable pattern, the location of failure was observed in the middle third close to the cervical third in 90% of the cases.

Saisho et al. (2023) compared the fracture strength of PEEK post-core, nanohybrid composite resin, polymer-infiltrated ceramic and fiber-reinforced epoxy resin post-core manufactured from CAD-CAM blocks using a method similar to that used in our study. The results of the study showed that the PEEK post-core was not statistically different from tooth-colored CAD-CAM milled post-core (Saisho et al., 2023). No statistical difference was found between the fracture strength values of PEEK post-cores produced by pressing and milling (Abdelmohsen et al., 2022).

Studies on post cores can be performed in vivo or in vitro. In vivo studies are difficult to perform and have high clinical importance. The aim of these studies was mostly to evaluate the validity of the results of in vitro studies. The biggest problems with such studies are that they are patient-dependent, and standardization cannot be achieved. In vitro studies are mostly used to test materials and techniques prior to clinical application. Therefore, mechanical tests that simulate intraoral conditions are important tools for evaluating restorative materials and techniques (De Oliveira et al., 2008; Nie et al., 2012). Although the results obtained from in vitro studies are thought to have little clinical significance, it is clear that they are helpful in determining clinical use protocols, especially when the findings obtained from well-designed in vivo studies in dentistry are considered. The controllability of variables in in vitro studies is convenient for standardization. In addition, in vitro studies are advantageous in terms of time and cost (Bayne, 2012).

In previous studies, acceptable results were obtained according to the results obtained from the 5-year follow-up of maxillary central teeth treated with custom PEEK posts. However, since there are limited clinical reports on the clinical applicability of PEEK posts, it is emphasized that long-term follow-up is needed (Kasem, Shams & Tribst, 2022).

PEEK posts have been compared with fiber posts, metal posts and aesthetic posts, which are most commonly used in routine treatments. In these studies, fracture strengths and failure modes of extracted teeth were evaluated by varying the materials, production methods and post sizes in previous experimental studies (Saisho et al., 2023; Abdelmohsen et al., 2022; Fathey, Azer & Abdelraheem, 2024; Ozarslan, US & Ozarslan, 2021). Our study is an original study in which custom modified PEEK posts (20% ceramic and 20% TiO2 filler) were compared with the most commonly used custom endodontic posts.

Our study had some limitations. Many factors significantly influence the results of in vitro fracture strength laboratory studies. The main factors are the design of the study, tooth selection, variable dimensions of the teeth, and the difficulty of simulating the intraoral environment. In addition, many factors, such as the classification of the tooth (anterior and posterior), degree of calcification, distance from the cemento-enamel junction of the area where the force is applied, direction of the force, and geometry of the casing are effective in determining the fracture strength of the extracted tooth. While the physical properties of the materials used in studies on post-cores determine their performance, factors that cannot be standardized within or between studies may lead to different results. Although similar-sized teeth were preferred, since the transversal cross-section of each tooth may be geometrically different, the preparation of step 1-mm in each region reflected this difference in the core base. Therefore, a standard mold cannot be used for the composite resin core.

CONCLUSIONS

In this study, the average fracture strength values of all groups were above the occlusal forces. Although there was no difference between the fracture strength values of PEEK post-core groups, zirconia and metal post-core groups showed the highest values. Repairable failure modes were more common than irreparable failure modes in the zirconia post-cores, whereas the opposite was observed in the other groups. However, more in-vitro, in-vivo and clinical follow-up studies are needed for the clinical use of PEEK as a post-core material.

Supplemental Information

Supplemental Information 1 Raw data

We are thankful to Fırat University Scientific Research Projects Center for the supply of materials and performing the experimental setup of this study (project number DHF.20.11).

Additional Information and Declarations

Competing Interests

Author Contributions

Human Ethics

Data Availability

The authors declare there are no competing interests.

Ayetullah Direk conceived and designed the experiments, performed the experiments, prepared figures and/or tables, authored or reviewed drafts of the article, and approved the final draft.

Samet Tekin conceived and designed the experiments, performed the experiments, analyzed the data, prepared figures and/or tables, authored or reviewed drafts of the article, and approved the final draft.

Zohaib Khurshid conceived and designed the experiments, analyzed the data, authored or reviewed drafts of the article, and approved the final draft.

The following information was supplied relating to ethical approvals (i.e., approving body and any reference numbers):

The Firat University granted Ethical approval from the Non-Interventional Research Ethics Committee on 2020/08-32.

The following information was supplied regarding data availability:

The raw data are available in the Supplemental Files.

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
