# Peer review of "Fracture strength of cad-cam milled polyetheretherketone (PEEK) post-cores vs conventional post-cores; an in vitro study"

_PeerJ, doi:10.7717/peerj.18012_

## Round 0.1 · original submission · Major Revisions

· Academic Editor

Major Revisions

Dear author
Please address point to point reviewers comments to improve your manuscript.

Reviewer 1 ·

Basic reporting

1. This manuscript needs revision to improve its quality.

2. The English language should be improved in grammar and writing style. I suggest the authors to send this manuscript for proofreading.

3. The introduction section needs more detail. I suggest the authors to discuss about fracture strength and fracture modes of the previous studies.

4. Please provide citations if necessary for example, the description in lines 80-87.

5. The study aim in the last paragraph of the introduction section is different from the aim you mentioned in the abstract. I suggest you to revise again this section to make it clearer for the audience.

Experimental design

1. In your methods, you mentioned that ‘written consent was obtained from eligible patients whose maxillary central incisors were planned to be extracted ….’, (lines 116-117) but in your study design on the next page, you were using mandibular premolars (line 121-122) as your samples. In the early statement of your abstract (line 24), you mentioned in the method section that you used mandibular first premolar as your sample. I suggest you to standardize the term that you use for your samples in this manuscript.

2. Why did you choose mandibular first premolar instead of maxillary central incisors?

3. How did you calculate the sample size? And how the 60 teeth were distributed among the 6 groups?

4. Who performed the RCT and who prepared the samples for this study? Did you send the samples to the dental technician or lab for preparation? If yes, how did you control this?

5. In line 201, you mentioned that ‘metal copings were prepared from Co-Cr metal alloy powder’, what is the purpose of preparing this metal coping? Did you mean the metal crown?

6. For the fracture mode test, it would be better if you could provide photos/images to show the fracture modes 1-5. It will provide a better illustration to the audience.

7. Please change the term study design to sample preparation or maybe you can make another subheading for sample preparation.

Validity of the findings

1. It would be better if the result of failure mode could be presented using photos/images instead of put in a table. Please discuss in detail the fracture modes in your discussion. For example, why did you choose stereomicroscope instead of scanning electron microscope?

2. I suggest you to improve in the discussion part. Please discuss more on your results.

3. You can state the limitations and clinical applicability of your study in the discussion. It is highly recommended.

Additional comments

No additional comments

Reviewer 2 ·

Basic reporting

The English language of the article needs improvement. I have made a few comments in the PDF, but it requires overall language enhancement. Additionally, more literature review is necessary, few relevant articles were not included.

Experimental design

The research question is well-defined. However, the author did not specify the number of experimental repetitions. It is unclear how many times the experiment was conducted or how the groups were statistically compared. Additional information on experimental repetitions and statistical analysis methods would enhance the clarity and reproducibility of the study.

Validity of the findings

The author did not mention how their work differs from that of others. It is unclear what the novelty of their research is. Additionally, the conclusions drawn in the abstract differ from those in the article. Clarifying the unique contributions of the study and ensuring consistency between the abstract and article conclusions would strengthen the quality.

Additional comments

More work is needed to improve the quality of the article.

Annotated reviews are not available for download in order to protect the identity of reviewers who chose to remain anonymous.

---

## Round 0.2 · Minor Revisions

· Academic Editor

Minor Revisions

Please revise line 134 to 138 and please make a last improvement on the English language.

Reviewer 1 ·

Basic reporting

No comment

Experimental design

1. Please revise again the sentence from line 134-138.

Validity of the findings

No comment

Additional comments

No additional comments

Reviewer 2 ·

Basic reporting

No comment

Experimental design

No comment

Validity of the findings

No comment

Additional comments

No comment

---

## Round 0.3 · accepted · Accept

· Academic Editor

Accept

All remaining concerns of the reviewers were adequately addressed and revised manuscript is acceptable now.